# Pilot Exploratory Study of a CT Radiomics Model for the Classification of Small Cell Lung Cancer and Non-Small-Cell Lung Cancer in the Moscow Population: A Step Toward Virtual Biopsy

**DOI:** 10.3390/jimaging11100331

**Published:** 2025-09-25

**Authors:** Maria D. Varyukhina, Alexandr A. Borisov, Rustam A. Erizhokov, Kirill M. Arzamasov, Alexander V. Solovev, Vadim V. Kirsanov, Olga V. Omelyanskaya, Anton V. Vladzymyrskyy, Yuriy A. Vasilev

**Affiliations:** 1Research and Practical Clinical Center for Diagnostics and Telemedicine Technologies of the Moscow Health Care Department, 127051 Moscow, Russia; borisovaa10@zdrav.mos.ru (A.A.B.); erizhokovra@zdrav.mos.ru (R.A.E.); arzamasovkm@zdrav.mos.ru (K.M.A.); solovevav10@zdrav.mos.ru (A.V.S.); kirsanovvv2@zdrav.mos.ru (V.V.K.); omelyanskayaov@zdrav.mos.ru (O.V.O.); vladzimirskijav@zdrav.mos.ru (A.V.V.); vasilevya1@zdrav.mos.ru (Y.A.V.); 2Department of Medical Cybernetics and Computer Science, Pirogov Russian National Research Medical University, 117513 Moscow, Russia

**Keywords:** lung cancer, radiomics, machine learning, small cell lung cancer, non-small cell lung cancer, computed tomography, tumor segmentation, diagnostic imaging

## Abstract

Lung cancer is one of the most common and socially significant cancers worldwide and consists of two main subtypes: small cell lung cancer (SCLC) and non-small cell lung cancer (NSCLC), which require different treatments. Computed tomography (CT) scans cannot reliably differentiate these subtypes, often necessitating invasive biopsies that carry significant risks. Radiomics offers a promising non-invasive alternative by quantitatively analyzing imaging data to extract detailed tissue characteristics beyond visual assessment. This pilot retrospective study analyzed 200 Moscow patients with histologically confirmed SCLC or NSCLC. Manual tumor segmentation on pretreatment CT scans allowed extraction of 107 radiomic features, from which 16 key features were selected to train four machine learning models. Models were evaluated using stratified 5-fold cross-validation, focusing on ROC AUC, accuracy, precision, and recall. All models demonstrated strong performance in distinguishing SCLC from NSCLC, with the gradient boosting model achieving the highest accuracy of 80.5% and ROC AUC of 0.888. These results highlight the potential of radiomics combined with machine learning to enable accurate, non-invasive differentiation of lung cancer subtypes. Further research is needed to expand feature sets, develop automated segmentation tools, and enhance clinical application of this approach.

## 1. Introduction

### 1.1. Scientific or Clinical Background

Lung cancer (LC) is one of the most common and socially significant oncological diseases worldwide. According to GLOBOCAN, in 2022, LC became the most frequently diagnosed type of cancer; 2.5 million new cases of lung cancer were registered (13.2% of the global cancer burden). There were also 1.8 million deaths (18.7% of all deaths) caused by LC, which makes it the leading cause of cancer-related deaths [1].Histologically, LC is divided into small cell cancer (SCLC) and non-small cell (NSCLC) subtypes, which differ significantly in the course of the disease, therapeutic approaches, and prognosis [2]. NSCLC is more common and proceeds more slowly, and SCLC is characterized by a lower prevalence but faster growth [3]. In 80% of patients with SCLC, the disease is detected in the presence of a metastatic lesion (stage IV or advanced form) and only in 20% with a localized form of stage I–III [4].Approaches to the treatment of various histological forms of lung cancer also differ significantly. According to the recommendations of the Russian Oncology Society, the main method of treating NSCLC is surgery. In the later stages, it is used in combination with chemotherapy, radiation therapy or chemoradiotherapy. Surgical treatment for SCLC is performed only for stage IA and stage IB in combination with mandatory adjuvant chemotherapy. The main treatment for SCLC is chemoradiotherapy [5].The main method of diagnosing lung cancer is computed tomography (CT) of the chest [6].

### 1.2. Rationale for Using a Radiomic Approach

However, standard CT of the chest, despite its high sensitivity in detecting formations (for nodes > 8 mm), does not reliably distinguish between SCLC and NSCLC due to overlapping radiological features such as indistinct contours, pleural cords, or central necrosis [7,8]. To determine the histological subtype of LC, doctors have to resort to invasive procedures such as percutaneous transthoracic lung biopsy under the control of computed tomography (PTLB), open lung biopsy (OLB) or a transbronchial biopsy. These manipulations have some limitations and are characterized by a high percentage of postmanipulatory complications. Thus, PTLB leads to pneumothorax in 20–25% of cases (2–15% of patients undergo thoracotomy), in 18% it leads to pulmonary hemorrhage, and in 4–5% of cases it leads to hemoptysis [9]. Up to 30% of patients undergoing OLB suffer biopsy-related complications such as pneumothorax, persistent air leaks, bleeding, and infections [10]. Transbronchial biopsy and cryo-transbronchial lung biopsy are possible only in the central form of lung cancer and are also associated with complications such as pneumothorax (6%), bleeding (2%) and bronchospasm or laryngospasm up to 2% [11]. Mortality in such procedures, despite its rarity (0.15%), remains a significant problem, especially in patients with concomitant chronic obstructive pulmonary disease or coagulopathies [12]. For some patients, it is impossible to perform an invasive diagnosis due to increased risks of perioperative morbidity and mortality from a combination of advanced age, comorbidity, severe respiratory failure, and pulmonary hypertension [13]. In this regard, it is important to develop and implement non-invasive techniques that can improve the accuracy of differential diagnosis of lung cancer.One of the promising approaches in this field is radiomics—mathematical analysis of medical radiation images that allows detecting tissue texture features at a level inaccessible to the eye of a radiologist [14]. Radiomics has found wide application in oncology, including for solving problems of differential diagnosis [15,16].

### 1.3. Study Objective

This pilot exploratory study aimed to develop and validate a radiomics-based machine learning model using preoperative CT scans to distinguish between SCLC and NSCLC subtypes, with histopathological diagnosis as the ground truth for the Moscow population.

## 2. Methods

### 2.1. Study Design

#### 2.1.1. Adherence to Guidelines or Checklists (e.g., CLEAR Checklist)

Our study followed the CLEAR checklist to support consistent and transparent reporting, addressing the need for standardized documentation in radiomics research [17].

#### 2.1.2. Ethical Details

This study was conducted in accordance with the Declaration of Helsinki and approved by the Independent Ethics Committee of Moscow Regional Branch of the Russian Society of Radiology (date: 19 June 2025; approval No.: 6).

#### 2.1.3. Sample Size Calculation

As a pilot exploratory study for the Moscow population, our sample size was determined by referencing similar studies, targeting 100 cases per class for binary classification, which, based on existing literature, is considered feasible for reliable radiomics analysis [18,19,20]. Also, the small sample size was associated with data availability due to low incidence of small cell lung cancer in the population, which is why we could not find more suitable cases for analysis.

#### 2.1.4. Study Nature (e.g., Retrospective, Prospective)

The current research was conducted as a non-interventional, retrospective analysis, utilizing pre-existing medical records and imaging data.

### 2.2. Eligibility Criteria

The study included adult patients (aged ≥18 years) diagnosed with SCLC or NSCLC, as classified under ICD-10 codes C33-C34 from 2019 to 2022, with pathologically confirmed diagnoses and available pretreatment computed tomography (CT) scans. Exclusion criteria encompassed CT scans of insufficient quality (e.g., exhibiting movement artifacts or obscured tumor boundaries due to peritumoral conditions such as atelectasis, metastases, inflammatory changes, pleural effusion, or hydrothorax) and cases with synchronous malignancies. Eligible cases were randomly selected from the Unified Radiological Information System of the Unified Medical Information and Analytical System of Moscow (ERIS-EMIAS) [21].

#### Flow for Technical Pipeline

Subsequent steps encompassed retrieval of pretreatment CT scans, tumor segmentation to isolate regions of interest (ROI), extraction of radiomic features, and binary classification of NSCLC vs. SCLC for which several commonly used machine-learning models were trained. Technical pipeline is presented in Figure 1.

### 2.3. Data

#### 2.3.1. Data Source (e.g., Private, Public)

Data for this study were sourced from the ERIS-EMIAS database, a centralized repository of medical imaging and clinical records of Moscow patients. No public data were used.

#### 2.3.2. Data Overlap

Cases were uniquely selected from the ERIS-EMIAS database, with each patient contributing a single pretreatment CT scan, thereby avoiding duplication or reuse of imaging data across the study cohort.

#### 2.3.3. Data Split Methodology

The dataset (*n* = 200 cases with 100 cases per class for binary classification) was divided into training and testing subsets using stratified 5-fold cross-validation with an 80:20 ratio. Cross-validation was selected to maximize data utilization, enhance model stability and generalizability, and minimize accidental errors associated with data splitting.

#### 2.3.4. Imaging Protocol

The study used native series of thin- slice CT scans of the chest with a slice thickness of 1.00 mm, pixel size 512 × 512 and a tube voltage of 120 kV (which was the most common scanning protocol in the population we evaluated). List of CT scanner models included in the study: Toshiba Aquilion 64 (Toshiba Corporation, Tokyo, Japan), Toshiba Aquilion CXL 128 (Toshiba Corporation, Tokyo, Japan), Toshiba Alexion TSX-032A (Toshiba Corporation, Tokyo, Japan), Toshiba Aquilion Prime (Toshiba Corporation, Tokyo, Japan), Siemens So-matom Definition (Siemens AG, Munich, Germany), Siemens Somatom Perspective (Sie-mens AG, Munich, Germany), Siemens Somatom Definition AS (Siemens AG, Munich, Germany), General Electric Revolution EVO (General Electric, Boston, MA, USA).

#### 2.3.5. Definition of Non-Radiomic Predictor Variables

No non-radiomic predictor variables, such as demographic characteristics, laboratory biomarkers, or traditional radiological assessments, were included in this study.

#### 2.3.6. Definition of the Reference Standard

The reference standard for this pilot exploratory study was the histopathological confirmation of lung cancer subtype (SCLC vs. NSCLC) from the cancer registry and the EMIAS. This outcome was chosen for its clinical relevance in guiding treatment and its high reproducibility via standardized registry data. The rationale was supported by prior studies on radiomics for SCLC vs. NSCLC classification. Potential reproducibility issues include the variability of histopathology between observers, which is eliminated with verified registration records, as well as potential bias in case selection, which is caused by the inclusion of many CT-scanned and confirmed histopathology cases, which limits the possibility of generalization.

### 2.4. Segmentation

#### 2.4.1. Segmentation Strategy

Segmentation was performed manually using 3D Slicer (version 5.8.1, https://www.slicer.org, accessed on 21 August 2025) on native pretreatment lung CT images [22]. A consensus-based approach was employed by radiologists, delineating tumor boundaries without margin shrinkage or expansion, including only the visible tumor contour and excluding peritumoral regions (e.g., atelectasis, pleural effusion) and cavities (decay cavities and bronchial lumens) (Figure 2). Default manual segmentation parameters (brush tool, no smoothing) were used. No image registration was required, as single-phase CT images were analyzed. Radiomic features were extracted from 3D tumor volumes. For patients with multiple lesions, only the primary tumor or the largest and most differentiated lesion from the surrounding tissues was segmented.

#### 2.4.2. Details of Operators Performing Segmentation

Segmentation was performed manually by two radiologists with over 5 years of experience in thoracic imaging, using the 3D Slicer. A consensus-based methodology was followed.

Our sample comprised an equal distribution (50:50) of patients with central and peripheral LC types. The central lesions proved to be more challenging to delineate from surrounding tissues on native CT series alone. This was reflected in a significantly lower inter-observer agreement for central lesions, as quantified by a Dice similarity coefficient (DSC) of 0.82 ± 0.13, compared to 0.93 ± 0.07 for peripheral lesions. To mitigate this variability, a consensus-based methodology was adopted, wherein only the regions of interest (ROI) identified by both radiologists were included in the final segmentation.

### 2.5. Pre-Processing

#### 2.5.1. Image Pre-Processing Details

No specialized preprocessing was applied to the CT scans.

#### 2.5.2. Resampling Method and Its Parameters

No resampling technique was applied to CT scans due to the use of a consistent imaging protocol with the same resolution.

#### 2.5.3. Discretization Method and Its Parameters

No manual discretization was applied for hand-crafted radiomic feature extraction, as features were automatically extracted using the Radiomics module in 3D Slicer.

#### 2.5.4. Image Types (e.g., Original, Filtered, Transformed)

Clinical images were used in their original clinical format, with only standard quality checks performed to exclude scans with artifacts or significant peritumoral changes.

### 2.6. Feature Extraction

#### 2.6.1. Feature Extraction Method

Hand-crafted radiomic features were extracted from segmented ROIs on pretreatment CT scans using the Radiomics module in 3D Slicer [23]. Radiomic feature dataset is provided in Appendix A.

#### 2.6.2. Feature Classes

Extracted radiomic features included first-order statistics, shape-based features and texture-based features.

#### 2.6.3. Number of Features

Overall number of the extracted features was 107.

#### 2.6.4. Default Configuration Statement for Remaining Parameters

All other parameters remained as a default configuration.

### 2.7. Data Preparation

#### 2.7.1. Handling of Missing Data

Patients without pretreatment CT scans or histopathological verification were excluded from the study. All included cases from the ERIS-EMIAS database had complete radiomic features extracted via 3DSlicer, with no missing values.

#### 2.7.2. Details of Class Imbalance

The dataset was balanced, with 100 cases for SCLC and NSCLC (including 50 cases for squamous cell carcinoma (SCC) and 50 cases for adenocarcinoma (ADC)).

#### 2.7.3. Details of Segmentation Reliability Analysis

No specific segmentation reliability analysis was conducted. A consensus-based approach was used to ensure agreement on tumor boundaries.

#### 2.7.4. Feature Scaling Details

Radiomic features, extracted using 3DSlicer with Z-score normalization applied to the entire CT image, were further standardized with StandardScaler for machine learning algorithms sensitive to scaling (logistic regression, support vector machine). For scale-invariant algorithms (random forest, gradient boosting), features were used without additional normalization.

#### 2.7.5. Dimension Reduction Details

Dimension reduction was performed with SmartCorrelatedSelection method from feature-engine library (https://feature-engine.trainindata.com, accessed on 21 August 2025). Feature groups with Spearman correlations exceeding 0.7 were identified and retained only the highest-importance feature from each group based on RandomForestClassifier’s ROC AUC performance evaluated through 3-fold cross-validation. Thus, 16 features from 107 were selected: [“Elongation”, “Flatness”, “Maximum2DDiameterSlice”, “Sphericity”, “90Percentile”, “Maximum”, “Skewness”, “Autocorrelation”, “DifferenceVariance”, “Imc1”, “InverseVariance”, “RunVariance”, “GrayLevelVariance.2”, “SmallAreaEmphasis”, “ZoneEntropy”, “Strength”].

### 2.8. Modeling

#### 2.8.1. Algorithm Details

Four machine learning algorithms were implemented using scikit-learn (v1.5.2, https://scikit-learn.org, accessed on 21 August 2025): random forest (tree ensemble), gradient boosting (sequential trees), logistic regression (linear), support vector machine (SVM). Inputs were 16 selected radiomic features; outputs were binary class probabilities. No architectural modifications or ensemble methods were applied.

#### 2.8.2. Training and Tuning Details

Models were trained on the training set with 5-fold cross-validation. The GridSearchCV class from the scikit-learn library (v1.5.2, https://scikit-learn.org, accessed on 21 August 2025) was used to optimize hyperparameters. Information about hyperparameters is presented in Table 1. No augmentation, transfer learning, or custom initialization was used. Training stopped when GridSearchCV identified optimal parameters.

#### 2.8.3. Handling of Confounders

No formal confounder detection or adjustment methods were applied. The cohort was balanced by sex between SCLC and NSCLC types and used a standardized imaging protocol, reducing concerns regarding these specific confounders.

#### 2.8.4. Model Selection Strategy

Machine learning models were assessed by performance metrics using 5-fold cross-validation. Model complexity was not constrained.

### 2.9. Evaluation

#### 2.9.1. Testing Technique (e.g., Internal, External)

No separate test set was used, instead, 5-fold cross-validation was performed.

#### 2.9.2. Performance Metrics and Rationale for Choosing

ROC AUC, accuracy, precision, and recall with 95% CIs were evaluated via 5-fold cross-validation.

#### 2.9.3. Uncertainty Evaluation and Measures (e.g., Confidence Intervals)

Uncertainty was quantified using 95% CIs for ROC AUC, accuracy, precision, and recall, calculated across 5-fold cross-validation folds. No additional robustness, sensitivity, or calibration analyses were performed in this study.

#### 2.9.4. Statistical Performance Comparison (e.g., DeLong’s Test)

No formal statistical comparison of model performance except 95% CIs calculation was conducted.

#### 2.9.5. Comparison with Non-Radiomic and Combined Methods

Descriptive comparison with combined clinical-radiomic methods is provided in Section 4.2.

#### 2.9.6. Interpretability and Explainability Methods

The SHapley Additive exPlanations (SHAP) method was adopted to interpret the predictions of the best-performing model.

## 3. Results

### 3.1. Baseline Demographic and Clinical Characteristic

Demographic characteristics of the patients are presented in Table 2. The patients across both lung cancer subtypes generally belong to a similar age group, and males predominate in both SCLC and NSCLC categories.

### 3.2. Flowchart for Eligibility Criteria

Patient records with ICD-10 lung cancer codes from the Russian Cancer Registry (2019–2022) were initially screened, totaling 3475 cases. From these, 530 patients with pathologically confirmed diagnoses were randomly selected from the ERIS-EMIAS database; after excluding those without pretreatment CT scans, poor-quality scans, obscured tumor boundaries, or synchronous malignancies, 200 patients remained, equally divided between SCLC and NSCLC subtypes. The eligibility criteria flowchart is presented in Figure 3.

### 3.3. Feature Statistics

Sixteen radiomic features were selected using an algorithmic approach, with features exhibiting correlations >0.7 excluded. The retained features spanned first-order, gray-level co-occurrence matrix (GLCM), gray-level run-length matrix (GLRLM), gray-level size-zone matrix (GLSZM), gray-level dependence matrix (GLDM), neighborhood gray-tone difference matrix (NGTDM), and shape classes. Table 3 lists the feature classes, names, and counts in the present study.

Feature importance was evaluated for Random Forest, Logistic Regression, SVM and Gradient Boosting models using the scikit-learn library with 5-fold stratified cross-validation. Results are presented in Figure 4.

To provide an in-depth explanation of the clinical significance of the features in the best-performing model (Gradient Boosting), SHAP analysis was performed. The bee swarm plot (Figure 5) quantified feature effects and confirmed that “GrayLevelVariance.2” was the most impactful feature, a finding consistent with the results of the aforementioned feature importance analysis.

### 3.4. Model Performance Evaluation

Performance metrics for Logistic Regression, Random Forest, Gradient Boosting, and SVM models were evaluated using 5-fold stratified cross-validation. Gradient Boosting achieved the highest ROC AUC (0.888, 95% CI: 0.839–0.938). ROC curves (Figure 6) illustrate model’s performance. Table 4 reports ROC AUC with 95% confidence intervals.

### 3.5. Comparison with Non-Radiomic and Combined Approaches

The descriptive comparison with combined clinical-radiomic methods is located in Section 4.2.

## 4. Discussion

### 4.1. Overview of Important Findings

The diagnostic accuracy metrics obtained confirm that radiomic features enable the differentiation of the two primary histological subtypes of lung cancer with reasonably high accuracy based on initial CT scan data. The Gradient Boosting model with an ROC AUC of 0.89 showed the best results.

According to the feature importance and SHAP analysis, the most significant feature for SCLC vs. NSCLC classification is the “GrayLevelVariance” of the GLDM. According to the values of this feature, SCLC is characterized by an average of 2 times higher variance of gray levels in the gray level dependence matrix of 45.9 [37.4; 53.1] versus 24.1 [7.5; 38.8] in NSCLC. This feature indicates a greater heterogeneity of small cell carcinomas. This feature separates squamous cell carcinomas with values of 7.8 [5.7; 13.6] especially well, which in the future can be used for deeper differentiation of histological types.

The second most important feature for the model is the “ZoneEntropy”, which measures the uncertainty or randomness in the distribution of zone sizes and gray levels. This feature also confirms the pattern demonstrated by the “GrayLevelVariance” feature: SCLC has the greatest heterogeneity of the textural pattern with values of 6.17 [6.03; 6.30], followed by ADC with values of 6.11 [5.95; 6.26] and SCC with values of 5.24 [4.87; 5.63] is the most homogeneous.

### 4.2. Previous Works with Differences from the Current Study

The issue of the use of radiomics in the diagnosis of lung cancer (LC) is increasingly covered in the scientific literature; however, studies focusing on the differential diagnosis of NSCLC and SCLC remain limited. For example, B.T. Chen et al. obtained an ROC AUC of 0.93 for the model of differentiation of ADC and SCLC using data from 69 patients [24]. J. Wang et al. developed a classification model for peripheral ADC and SCLC based on a combination of clinical, radiological, and radiomics data from 240 patients. This model showed an ROC AUC of 0.967 in the validation sample [25]. Despite its strong performance, a significant drawback of this study is also the absence of SCC cases, which is one of the most common LC subtypes. S. Liu et al.’s work uses data from 468 patients with NSCLC and SCLC, but it lacks the information about histological types of NSCLC. Their model demonstrated an ROC AUC of 0.86 (95% CI: 0.82–0.90) in the training set and 0.82 (95% CI: 0.75–0.89) in the validation set [26]. Collectively, these works highlight a clear need for more comprehensive studies that include the major NSCLC subtypes to develop robust and clinically applicable differential diagnostic models.

The study by H. Li et al. directly addressed this gap by using data from 200 patients with ADC, SCC and SCLC. Their methodology, however, relied on pairwise classification models rather than a unified multi-class framework. The highest ROC AUCs in classifying ADC vs. SCC, ADC vs. SCLC, and SCC vs. SCLC were 0.879, 0.836, 0.783, respectively, by using a feedforward neural network [27].

R.P. Shah et al. proposed a noteworthy approach for differentiating SCLC from other LC types and benign tumors. Their model, trained on a cohort of 103 patients, achieved its best performance (ROC AUC = 0.87) by integrating radiomic features from both native and contrast-enhanced CT series [28].

Borisov et al. developed a classification model to differentiate neuroendocrine neoplasms from NSCLC with an achieved AUC = 0.98. This strong performance proves evidence of the underlying differences in the radiological phenotypes of these tumor types [29].

Numerous studies have demonstrated that integrating clinical data with radiomic features can significantly enhance the performance of models for LC subclassification. Therefore, T. Zhang et al. used serum tumor markers to differentiate ADC and SCC. Their model achieved an AUC = 0.921 on the validation set [30]. S. Liu et al. have developed a combined model that includes radiomic features, age, sex, smoking history, clinical stage of LC, maximum tumor diameter, and serum tumor markers. This model demonstrated AUC = 0.94 and exceeded the values of the clinical and radiomic models [26]. Chen BT et al. also noted that the addition of clinical features to the model allowed for a slight improvement in its AUC [24]. Thus, the addition of clinical and laboratory features can enhance the capabilities of lung cancer differentiation models. Future research will therefore prioritize evaluating the potential of such multimodal approaches to improve the accuracy and robustness of LC subtype differentiation.

Within the context of solely radiomics-based models, our study is most similar to the work of E. Linning et al., where, based on native CT scans of 278 patients, classification models of NSCLC and SCLC, ADC and SCC were trained, with ROC AUC of 0.741 and 0.655, respectively, which is lower than our results [31].

### 4.3. Practical Implications

Potentially, upon further validation, our model could be applied to differentiate the type of intrapulmonary tumor detected on non-contrast chest CT. This may be particularly relevant in clinical scenarios where tumor biopsy is impossible, carries high risks, or yields ambiguous cytological results. However, a formal assessment of the model’s clinical utility—evaluating its net benefit over existing clinical decision pathways and its impact on patient outcomes—is a necessary next step before any routine clinical application can be considered.

A key obstacle to its clinical integration is the current necessity for manual tumor segmentation, which is time-consuming and challenging in routine practice. A future direction to overcome this limitation is the development of an automated segmentation tool, although this presents a significant technical challenge, especially for central tumors poorly differentiated from surrounding tissues.

The predictive power of our model could be further enhanced by expanding the set of radiomic features, for instance, by incorporating wavelet transforms and varying Laplacian of Gaussian (LoG) kernel sizes. Additionally, integrating clinically relevant variables—such as patient demographics, smoking history, serum tumor markers or imaging characteristics beyond the tumor region—could significantly improve the model’s discriminatory ability and clinical relevance. Furthermore, increasing the sample size and including rarer histological subtypes of lung cancer will be essential to improve the model’s robustness and generalizability.

### 4.4. Strengths and Limitations

A key advantage of our work is the consensus-based segmentation process, which minimizes the risk of human error during lesion markup, which is critically important for central types of LC.

Another strong point of our work is the unification of the scanning parameters used in the research. Although this approach reduces the number of available examinations, the reproducibility of the results obtained increases without complicating the preprocessing processes of the examination.

A primary limitation of this study is the exclusive use of non-contrast CT series. While the integration of contrast-enhanced CT is well recognized as a promising strategy for improving segmentation accuracy and radiomic feature robustness, it was deliberately excluded. This decision was based on its infrequent use during the initial diagnostic workup for lung cancer in the relevant clinical setting. Furthermore, a significant portion of the patient cohort did not undergo contrast-enhanced CT prior to treatment. Its inclusion would have substantially reduced the sample size of this pilot investigation. Therefore, the evaluation of contrast-enhanced CT remains a critical objective for future research.

The study was intentionally limited to radiomic features, and thus no non-radiomic variables (e.g., age, sex) were included in the model. It should be noted that the cohort was balanced by sex between SCLC and NSCLC types and used a standardized imaging protocol, reducing concerns regarding these specific confounders. The effect of age remains unanalyzed and is a candidate for future investigation alongside other clinical characteristics. As evaluating technical confounders (e.g., scanner variability) was not an objective, they were not formally analyzed, and low-quality scans were excluded to ensure reliability.

While the present study has focused on distinguishing between the two broad categories of lung cancer (SCLC and NSCLC), the broader potential of radiomics lies in its ability to differentiate between specific histological subtypes. Therefore, future work will be directed toward developing models to discriminate between entities such as SCLC and ADC, SCLC and SCC, and ADC vs. SCC.

Previous studies indicate a trend toward higher accuracy in models that combine radiomic features with clinical data. As the present analysis focused specifically on evaluating the predictive power of radiomics alone, subsequent investigations will incorporate relevant clinical variables to enhance classification performance.

This research represents an early-stage investigation focused primarily on technical validation. Consequently, the assessment of the model’s clinical utility, for instance through Decision-Curve Analysis (DCA), was beyond the scope of the present study. A formal evaluation of clinical net benefit remains an essential next step for translating the model into potential clinical practice and will be a critical focus of subsequent work.

This work constitutes a pilot study in the Moscow population, demonstrating the feasibility of using radiomic features to differentiate LC types within the operational framework of the local radiology service. Although the Moscow population is substantial in size, the risk of population bias remains. Therefore, future research should prioritize external validation to evaluate generalizability across populations and imaging protocols. Additionally, efforts to enhance the reproducibility of the entire radiomic pipeline are essential to ensure robust and translatable results.

## Figures and Tables

**Figure 1 jimaging-11-00331-f001:**
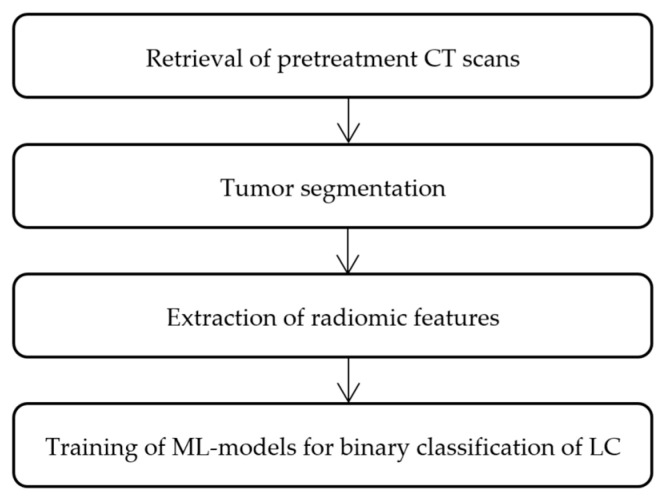
Flowchart for technical pipeline of the current radiomics analysis.

**Figure 2 jimaging-11-00331-f002:**
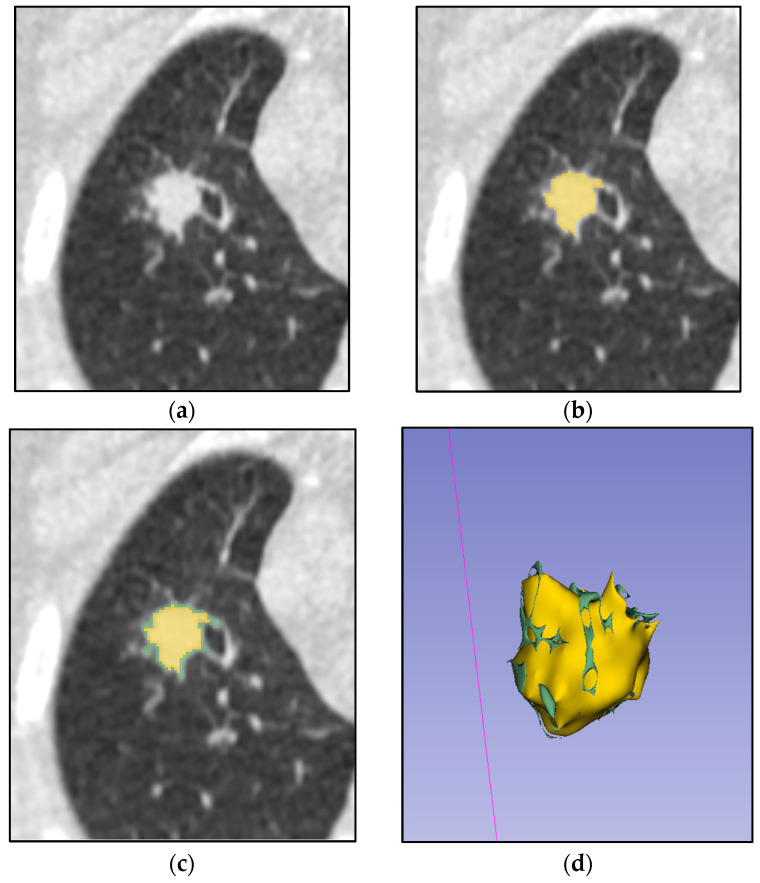
Process of manual tumor segmentation by two independent radiologists. (**a**) Native CT image of a lung with tumor; (**b**) ROI (tumor) segmented by one expert (yellow mask); (**c**) Combined mask of two experts’ segmentation (joined yellow and green masks); (**d**) 3D model of two experts’ combined segmentation.

**Figure 3 jimaging-11-00331-f003:**
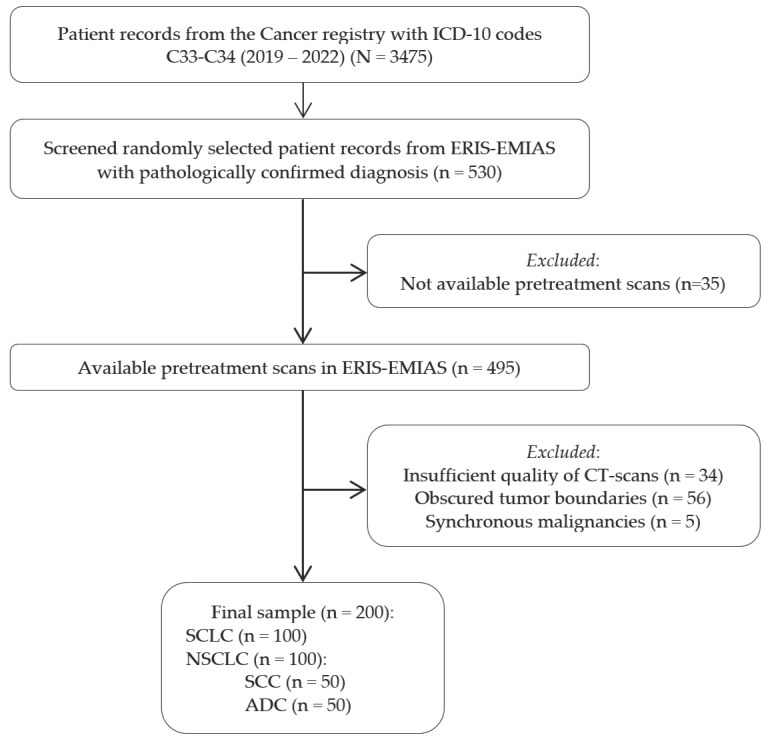
Eligibility criteria flowchart illustrating the patient selection process from the Russian Cancer Registry and ERIS-EMIAS database, showing inclusion and exclusion criteria that resulted in the final sample of 200 lung cancer patients.

**Figure 4 jimaging-11-00331-f004:**
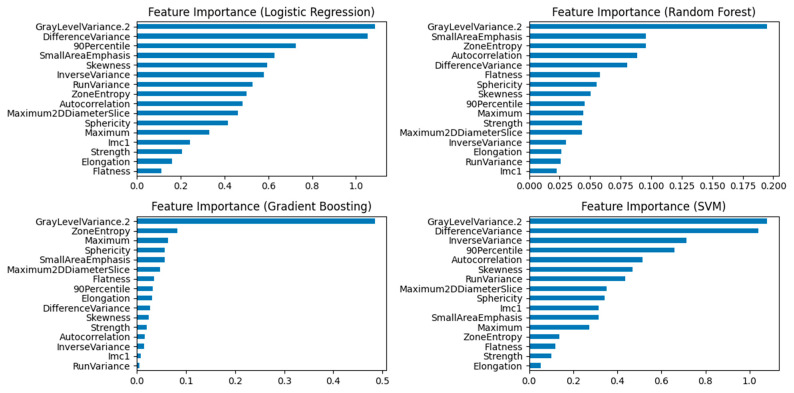
Feature Importance plots for Logistic Regression, Random Forest, Gradient Boosting, and SVM Models. The “GrayLevelVariance.2” feature emerged as the leading contributor across all models.

**Figure 5 jimaging-11-00331-f005:**
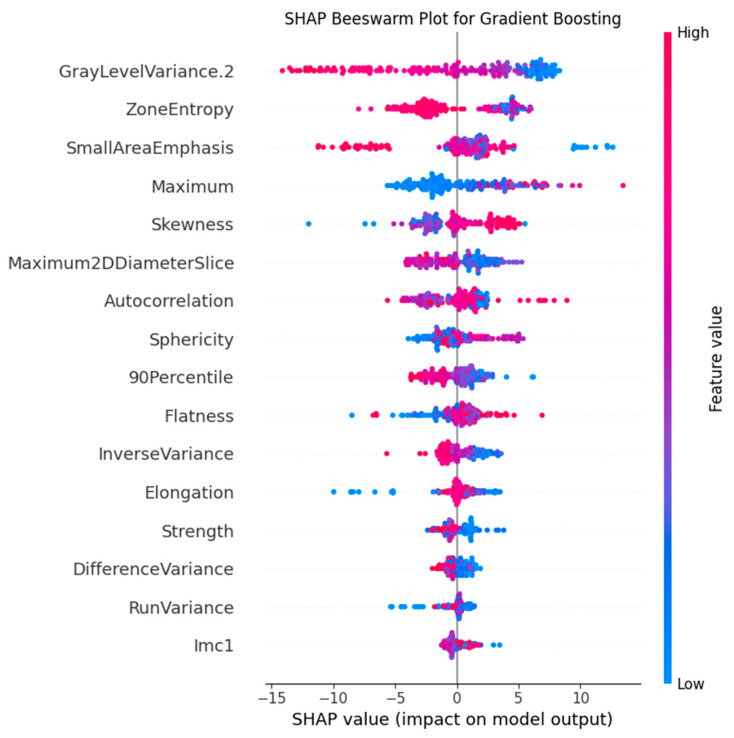
SHAP plot for the Gradient Boosting model. The plot illustrates the contribution of the top predictive features, with “GrayLevelVariance.2” identified as the most influential.

**Figure 6 jimaging-11-00331-f006:**
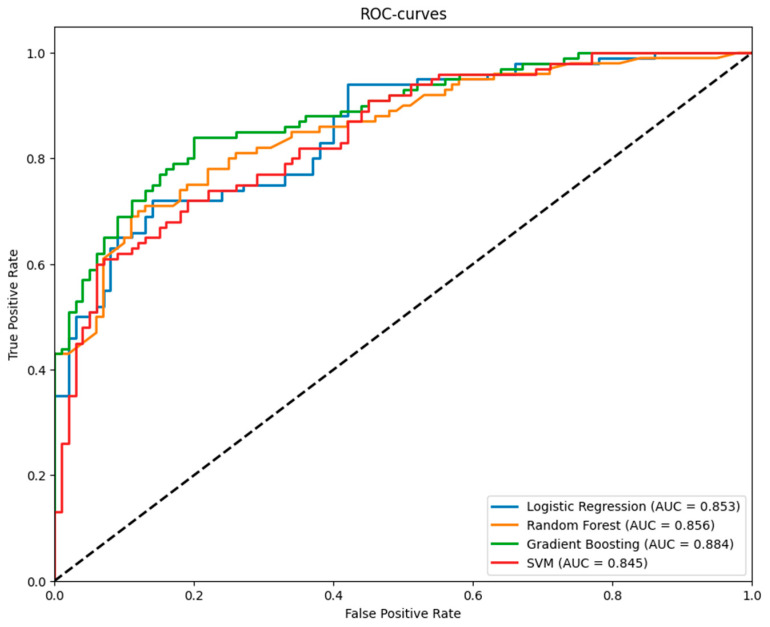
ROC Curves for Logistic Regression, Random Forest, Gradient Boosting, and Support Vector Machine Models from 5-Fold Cross-Validation.

**Table 1 jimaging-11-00331-t001:** Information about the hyperparameter grid used for optimization for each model.

Model	GridSearchCV Hyperparameters Grid
Logistic Regression	C: [0.001, 0.01, 0.1, 1, 10, 100], penalty: [11, 12]
Random Forest	n_estimators: [50, 100, 200], max_depth: [None, 5, 10], min_samples_split: [2, 5, 10]
Gradient Boosting	n_estimators: [50, 100, 200], learning_rate: [0.01, 0.1, 0.5], max_depth: [3, 5, 7]
SVM	C: [0.1, 1, 10], kernel: [linear, rbf], gamma: [scale, auto]

**Table 2 jimaging-11-00331-t002:** Distribution of patients with SCLC and NSCLC, including adenocarcinoma and squamous cell carcinoma subtypes, by sex, count, and age statistics.

Cancer Type	Histological Type	Sex	Count	Mean Age	Std	Min Age	25%	50%	75%	Max Age
NSCLC	Adenocarcinoma	F	17	71.4	10.1	53.0	65.0	73.0	78.0	93.0
		M	33	69.5	8.3	51.0	66.0	70.0	74.0	94.0
	Squamous cell cancer	F	4	60.8	9.9	48.0	57.6	61.5	64.5	72.0
		M	46	69.5	7.4	53.0	65.3	69.5	74.0	89.0
	Total	F	21	69.4	10.7	48.0	62.0	72.0	74.0	93.0
		M	79	69.5	7.7	51.0	66.0	70.0	74.0	94.0
SCLC	Small cell lung cancer	F	21	65.5	11.4	37.0	60.0	67.0	72.0	80.0
		M	79	65.9	7.3	40.0	62.0	66.0	71.0	85.0

**Table 3 jimaging-11-00331-t003:** Selected radiomic features.

Feature Class	Feature Names	N of Features
First order	90Percentile, Maximum, Skewness	3
GLCM	DifferenceVariance, InverseVariance, Autocorrelation, Imc1	4
GLRLM	RunVariance	1
GLSZM	SmallAreaEmphasis, ZoneEntropy	2
GLDM	GrayLevelVariance.2	1
NGTDM	Strength	1
Shape	Maximum2DDiameterSlice, Sphericity, Elongation, Flatness	4
Total		16

**Table 4 jimaging-11-00331-t004:** ROC AUC with 95% Confidence Intervals for Logistic Regression, Random Forest, Gradient Boosting, and Support Vector Machine Models from 5-Fold Cross-Validation. Gradient boosting showed the best results in ROC AUC, accuracy, and recall metrics.

Model	ROC AUC (95% CI)	Accuracy (95% CI)	Precision (95% CI)	Recall (95% CI)
Logistic Regression	0.853 (0.772–0.934)	0.750 (0.662–0.838)	0.769 (0.673–0.864)	0.720 (0.584–0.856)
Random Forest	0.856 (0.782–0.930)	0.775 (0.702–0.848)	0.824 (0.651–0.998)	0.740 (0.613–0.867)
Gradient Boosting	0.888 (0.839–0.938)	0.805 (0.745–0.865)	0.816 (0.721–0.911)	0.800 (0.702–0.898)
SVM	0.847 (0.783–0.910)	0.745 (0.665–0.825)	0.775 (0.636–0.914)	0.720 (0.616–0.824)

## Data Availability

The original contributions presented in this study are included in the article/Appendix A. Further inquiries can be directed to the corresponding author.

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
