# Peer review of "Pilot Exploratory Study of a CT Radiomics Model for the Classification of Small Cell Lung Cancer and Non-Small-Cell Lung Cancer in the Moscow Population: A Step Toward Virtual Biopsy"

_2313-433X, 2025, doi:10.3390/jimaging11100331_

Round 1

Reviewer 1 Report

Comments and Suggestions for Authors

In this research paper(jimaging-3861088), this study explores the potential of a CT radiomics model combined with machine learning to non-invasively distinguish between small cell lung cancer (SCLC) and non-small cell lung cancer (NSCLC) in a Moscow population, achieving promising results with the gradient boosting model demonstrating the highest accuracy and ROC AUC. I would like to recommend this manuscript to be published in Journal of Imaging after major revisions.

Some detail information need to be clarified :

  1. The study did not conduct reliability analysis of the segmentation results, such as calculating the Dice similarity coefficient. It is suggested to supplement this analysis in the Methods section.
  2. The study used only non - contrast - enhanced CT images for feature extraction without considering multiphase or contrast - enhanced images. It is suggested to mention in the Discussion section the additional information that these images may provide.
  3. The study lacks an assessment of the model's clinical utility, such as decision - curve analysis (DCA). It is suggested to mention in the Discussion section the importance of this assessment.
  4. In the Discussion section, a subsection titled "Comparison with Existing Studies" could be added to provide a detailed analysis of the similarities and differences between this study and other similar research.

Author Response

Dear Reviewer,

Thank you for your thorough review of our manuscript and for your valuable comments. Your feedback has been immensely helpful in improving our work. We have carefully addressed all the points you raised and have incorporated the necessary revisions into the manuscript. Below, we provide a point-by-point response to each of your comments, detailing the changes made.

Comment 1: “The study did not conduct reliability analysis of the segmentation results, such as calculating the Dice similarity coefficient. It is suggested to supplement this analysis in the Methods section.”

Reply to Comment 1: Thank you for this important suggestion. We agree that assessing the inter-observer reliability of tumor segmentation is crucial for ensuring the robustness of radiomic feature extraction. As recommended, we have included an analysis of segmentation consistency.

Changes in the text: in the Methods section (subsection "2.4.2 Details of operators performing segmentation") added text “Our sample comprised an equal distribution (50:50) of patients with central and peripheral LC types. The central lesions proved to be more challenging to delineate from surrounding tissues on native CT series alone. This was reflected in a significantly lower inter-observer agreement for central lesions, as quantified by a Dice similarity coefficient (DSC) of 0.82 ± 0.13, compared to 0.93 ± 0.07 for peripheral lesions. To mitigate this variability, a consensus-based methodology was adopted, wherein only the regions of interest (ROI) identified by both radiologists were included in the final segmentation.”

Comment 2: “The study used only non - contrast - enhanced CT images for feature extraction without considering multiphase or contrast - enhanced images. It is suggested to mention in the Discussion section the additional information that these images may provide.”

Reply to Comment 2: Thank you for raising this valuable point. We agree that multiphase or contrast-enhanced CT images can provide critical additional information about key characteristics for differentiating lung cancer subtypes. The primary reason for non-including contrast-enhanced series in this analysis was their lack of availability for a significant portion of the patient cohort. Including only patients with contrast-enhanced CT would have substantially narrowed the sample size and compromised the statistical power of this pilot investigation.

Changes in the text: in the Discussion section (subsection “4.4 Strengths and limitations”) added text “A primary limitation of this study is the exclusive use of non-contrast CT series. While the integration of contrast-enhanced CT is well recognized as a promising strategy for improving segmentation accuracy and radiomic feature robustness, it was deliberately excluded. This decision was based on its infrequent use during the initial diagnostic workup for lung cancer in the relevant clinical setting. Furthermore, a significant portion of the patient cohort did not undergo contrast-enhanced CT prior to treatment. Its inclusion would have substantially reduced the sample size of this pilot investigation. Therefore, the evaluation of contrast-enhanced CT remains a critical objective for future research.”

Comment 3: “The study lacks an assessment of the model's clinical utility, such as decision - curve analysis (DCA). It is suggested to mention in the Discussion section the importance of this assessment.”

Reply to Comment 3: We thank the reviewer for this valuable comment. We have now addressed it in the Discussion by acknowledging the need for DCA to assess clinical utility in future work (sections 4.3 and 4.4).

Changes in the text:

In the Discussion section (subsection “4.3 Practical implications”) added text “However, a formal assessment of the model's clinical utility—evaluating its net benefit over existing clinical decision pathways and its impact on patient outcomes—is a necessary next step before any routine clinical application can be considered.”

In the Discussion section (subsection “4.4 Strengths and limitations”) added text “This research represents an early-stage investigation focused primarily on technical validation. Consequently, the assessment of the model's clinical utility, for instance through Decision-Curve Analysis (DCA), was beyond the scope of the present study. A formal evaluation of clinical net benefit remains an essential next step for translating the model into potential clinical practice and will be a critical focus of subsequent work.”

Comment 4: “In the Discussion section, a subsection titled "Comparison with Existing Studies" could be added to provide a detailed analysis of the similarities and differences between this study and other similar research.”

Reply to Comment 4: Thank you for this valuable suggestion. We have added a dedicated paragraph in the Discussion section.

Changes in the text:

In the Discussion section (subsection “4.2 Previous works with differences from the current study”) added text “Numerous studies have demonstrated that integrating clinical data with radiomic features can significantly enhance the performance of models for LC subclassification. So T. Zhang et al. used serum tumor markers to differentiate ADC and SCC. Their model achieved AUC = 0.921 on validation set [30]. S. Liu et al. have developed a combined mod-el that includes radiomic features, age, sex, smoking history, clinical stage of LC, maximum tumor diameter, and serum tumor markers. This model demonstrated AUC = 0.94 and exceeded the values of the clinical and radiomic models [26]. Chen BT et al. also noted that the addition of clinical features to the model allowed for a slight improvement in its AUC [24]. Thus, the addition of clinical and laboratory features can enhance the capabilities of lung cancer differentiation models. Future research will therefore prioritize evaluating the potential of such multimodal approaches to improve the accuracy and robustness of LC subtype differentiation.”

We thank the Reviewer for the thorough and constructive feedback, which has significantly improved the quality of our manuscript. In addition to the specific revisions outlined above, we have performed a comprehensive stylistic editing of the text, incorporated a feature interpretability analysis using SHAP, shared our dataset with radiomic features and enhanced the visual content (figures and tables) for greater clarity. All comments from the other reviewers have also been carefully addressed. We believe the manuscript is now substantially stronger and hope it is deemed suitable for publication.

Reviewer 2 Report

Comments and Suggestions for Authors

This pilot exploratory study presents a radiomics-based machine learning approach to differentiate between SCLC and NSCLC using CT scans from a Moscow-based population. The topic is clinically relevant, and the methodology is generally sound. The study demonstrates promising results, with the gradient boosting model achieving an AUC of 0.888. However, several limitations should be addressed to strengthen the manuscript.

  1. Inter-observer consistency analysis (such as Dice coefficient) was not provided and should be supplemented to ensure reproducibility.
  2. Clinical variables (such as age, smoking history) were not integrated, nor was a comparison made with traditional imaging assessments. Incorporating clinical features might improve model performance.
  3. Although feature importance was provided, there was a lack of in-depth explanation of the clinical significance of the features. It is recommended to introduce interpretability methods such as SHAP.
  4. Although the study mentioned that NSCLC includes SCC and ADC, subgroup analyses specifically for SCLC vs. SCC and SCLC vs. ADC were not conducted. This makes it impossible to clarify the model's performance differences in distinguishing between SCLC and different NSCLC subtypes. It is recommended to add subgroup analyses in subsequent work, calculating metrics like accuracy and ROC AUC for distinguishing SCLC from SCC and SCLC from ADC, and analyzing the model's diagnostic efficacy across different subtype combinations to provide reference for more precise lung cancer subtype diagnosis in clinical practice.
  5. Data or code were not made public, which affects reproducibility. It is recommended to share anonymized feature data or code within permissible limits.
  6. The methods section stated that ethical approval was not required, but the latter part mentioned approval from an ethics committee. The statements need to be unified.
  7. The sample size was relatively small (n=200) and came from a single region. It is recommended that subsequent studies include external validation (e.g., incorporating independent datasets from other cities or countries) to enhance generalizability.

Author Response

Dear Reviewer,

We sincerely thank the reviewer for the thorough review and highly valuable comments, which have greatly improved our manuscript. All raised points have been carefully addressed, and the corresponding revisions have been incorporated. Our detailed point-by-point responses are provided below.

Comment 1: “Inter-observer consistency analysis (such as Dice coefficient) was not provided and should be supplemented to ensure reproducibility.”

Reply to Comment 1: Thank you for raising this valuable point. We have included an analysis of segmentation consistency.

Changes in the text: in the Methods section (subsection "2.4.2 Details of operators performing segmentation") added text “Our sample comprised an equal distribution (50:50) of patients with central and peripheral LC types. The central lesions proved to be more challenging to delineate from surrounding tissues on native CT series alone. This was reflected in a significantly lower inter-observer agreement for central lesions, as quantified by a Dice similarity coefficient (DSC) of 0.82 ± 0.13, compared to 0.93 ± 0.07 for peripheral lesions. To mitigate this variability, a consensus-based methodology was adopted, wherein only the regions of interest (ROI) identified by both radiologists were included in the final segmentation.”

Comment 2: “Clinical variables (such as age, smoking history) were not integrated, nor was a comparison made with traditional imaging assessments. Incorporating clinical features might improve model performance.”

Reply to Comment 4: Thank you for this valuable suggestion. We have added dedicated paragraphs in the Discussion section to provide comparison with existing works, describe its clinical utility as well as highlight limitations of our research.

Changes in the text:

In the Discussion section (subsection “4.2 Previous works with differences from the current study”) added text “Numerous studies have demonstrated that integrating clinical data with radiomic features can significantly enhance the performance of models for LC subclassification. So T. Zhang et al. used serum tumor markers to differentiate ADC and SCC. Their model achieved AUC = 0.921 on validation set [30]. S. Liu et al. have developed a combined mod-el that includes radiomic features, age, sex, smoking history, clinical stage of LC, maximum tumor diameter, and serum tumor markers. This model demonstrated AUC = 0.94 and exceeded the values of the clinical and radiomic models [26]. Chen BT et al. also noted that the addition of clinical features to the model allowed for a slight improvement in its AUC [24]. Thus, the addition of clinical and laboratory features can enhance the capabilities of lung cancer differentiation models. Future research will therefore prioritize evaluating the potential of such multimodal approaches to improve the accuracy and robustness of LC subtype differentiation.”

In the Discussion section (subsection “4.3 Practical implications”) added text “Additionally, integrating clinically relevant variables—such as patient demographics, smoking history, serum tumor markers or imaging characteristics beyond the tumor region—could significantly improve the model's discriminatory ability and clinical relevance.”

In the Discussion section (subsection 4.4 Strength and limitations) added text “The study was intentionally limited to radiomic features, and thus no non-radiomic variables (e.g., age, sex) were included in the model. It should be noted that the cohort was balanced by sex between SCLC and NSCLC types and used a standardized imaging protocol, reducing concerns regarding these specific confounders. The effect of age remains unanalyzed and is a candidate for future investigation alongside other clinical characteristics. As evaluating technical confounders (e.g., scanner variability) was not an objective, they were not formally analyzed, and low-quality scans were excluded to ensure reliability.”

Comment 3: “Although feature importance was provided, there was a lack of in-depth explanation of the clinical significance of the features. It is recommended to introduce interpretability methods such as SHAP.”

Reply to Comment 3: Thank you for this valuable recommendation. We have added results of SHAP analysis in the Results section and slightly changed the Discussion section.

Changes in the text: In the Results section (subsection “3.3 Feature statistics”) added text “To provide an in-depth explanation of the clinical significance of the features in the best-performing model (Gradient Boosting), SHAP analysis was performed. The beeswarm plot (Figure 5) quantified feature effects and confirmed that “GrayLevelVariance.2” was the most impactful feature, a finding consistent with the results of the afore-mentioned feature importance analysis.” and added Figure 5, representing SHAP plot for the Gradient Boosting model (the best performed one). The plot illustrates the contribution of the top predictive features, with "GrayLevelVariance.2" identified as the most influential.

In the Discussion section (subsection “4.1 Overview of important findings”) added text “According the feature importance and SHAP analysis, the most significant feature for SCLC vs NSCLC classification is the “GrayLevelVariance” of the GLDM matrix.”

Comment 4: “Although the study mentioned that NSCLC includes SCC and ADC, subgroup analyses specifically for SCLC vs. SCC and SCLC vs. ADC were not conducted. This makes it impossible to clarify the model's performance differences in distinguishing between SCLC and different NSCLC subtypes. It is recommended to add subgroup analyses in subsequent work, calculating metrics like accuracy and ROC AUC for distinguishing SCLC from SCC and SCLC from ADC, and analyzing the model's diagnostic efficacy across different subtype combinations to provide reference for more precise lung cancer subtype diagnosis in clinical practice.”

Reply to Comment 4: We thank the reviewer for this valuable comment. We have broadened the limitations of our study and highlighted the importance of this analysis in further research.

Changes in the text: In the Discussion section (subsection “4.4 Strengths and limitations”) added text “While the present study has focused on distinguishing between the two broad cate-gories of lung cancer (SCLC and NSCLC), the broader potential of radiomics lies in its ability to differentiate between specific histological subtypes. Therefore, future work will be directed toward developing models to discriminate between entities such as SCLC and ADC, SCLC and SCC, and ADC vs SCC.”

Comment 5: “Data or code were not made public, which affects reproducibility. It is recommended to share anonymized feature data or code within permissible limits.”

Reply to Comment 5: Thank you for raising this important point regarding reproducibility. In response to your comment, we have made the radiomic feature dataset used in this study publicly available as a supplementary file accompanying this article. We agree that this will facilitate the validation and replication of our findings.

Changes in the text: In the Data Availability Statement section added text “The radiomic feature dataset formed and used in this study is included in this published article (and its supplementary information files). Due to institutional policies, the raw medical images, segmentation data, preprocessing scripts, source code, model files, and executable systems are not publicly available.”

Comment 6: “The methods section stated that ethical approval was not required, but the latter part mentioned approval from an ethics committee. The statements need to be unified.”

Reply to Comment 6: Thank you for catching this discrepancy. It was indeed a typographical error. The text has been corrected.

Changes in the text: In the Methods section (subsection “2.1.2 Ethical details”) added text “This study was conducted in accordance with the Declaration of Helsinki, and approved by the Independent Ethics Committee of Moscow Regional Branch of the Russian Society of Radiology (date: 19 June 2025; approval â„–: 6)”.

Comment 7: “The sample size was relatively small (n=200) and came from a single region. It is recommended that subsequent studies include external validation (e.g., incorporating independent datasets from other cities or countries) to enhance generalizability.”

Reply to Comment 7: We thank the reviewer for this comment. We have broadened the limitations of our study and highlighted the importance of external validation in further research.

Changes in the text: In the Discussion section (subsection “4.4 Strengths and limitations”) added text “This work constitutes a pilot study in the Moscow population, demonstrating the feasibility of using radiomic features to differentiate LC types within the operational framework of the local radiology service. Although the Moscow population is substantial in size, the risk of population bias remains. Therefore, future research should prioritize external validation to evaluate generalizability across populations and imaging protocols. Additionally, efforts to enhance the reproducibility of the entire radiomic pipeline are essential to ensure robust and translatable results.”

We thank the Reviewer for their thorough and constructive feedback, which has greatly enhanced our manuscript. Beyond the specific revisions detailed in our responses, we have conducted a comprehensive stylistic edit of the text and improved the visual content (figures and tables) to ensure greater clarity. All comments from the co-reviewers have been diligently incorporated as well. We are confident that these revisions have strengthened the manuscript considerably and hope that it now meets the journal's standards for publication.

Reviewer 3 Report

Comments and Suggestions for Authors

Following a thorough examination of the article, the ensuing discussion will address its salient strengths.
The topic is both relevant and current. The non-invasive differentiation of small cell lung cancer (SCLC) and non-small cell lung cancer (NSCLC) using radiomics is an emerging field with high clinical value. The methodology employed is clear. The pipeline (segmentation, feature extraction, selection, modelling) is described in detail. The attainment of consistent results is of paramount importance. The Gradient Boosting algorithm attained an Area Under the Curve (AUC) value of 0.888, a result that can be considered satisfactory for the context of a preliminary study. Bibliographic context: The discussion draws parallels with several previous studies, thereby demonstrating a comprehensive understanding of the current state of research. The structure is evidently well-organised and adheres to the stipulated guidelines (CLEAR).
The following essay will provide a comprehensive overview of the relevant literature on the subject.

Conversely, the following limitations are apparent: The following aspects of methodology require further discussion:
The sample size of 200 cases is limited, and although the study is justified as a pilot, the statistical power and risk of overfitting should be discussed in more detail. The following essay will provide a comprehensive overview of the relevant literature on the subject.
Validation: It should be noted that the present study employed internal validation techniques exclusively, namely 5-fold cross-validation. The absence of external set or multicentre validation results in a reduction of generalisation. Manual segmentation: Despite the utilisation of a consensus of two radiologists, no interobserver reliability analysis (e.g., kappa, DSC) was conducted. This represents a significant limitation in the field of radiomics. Absence of confounder analysis: The influence of variables such as age, sex, or variability between scans was not studied. Interpretability: The application of explainability techniques (SHAP, LIME, more detailed feature importance) was not implemented, thereby restricting the clinical interpretability of the models. In terms of the clarity of the writing, it is submitted that some sections are very descriptive (e.g. '2.3 Data' or '2.4 Segmentation') and could be summarised, with the technical details being relegated to an appendix.
The following essay will provide a comprehensive overview of the relevant literature on the subject.

The English is correct, but not very fluent in places. For example, the phrase 'in the study used native series' should be written as 'the study used native series'. The text requires linguistic revision for the purpose of international publication.
It is evident that there is a redundancy in the repeated mention of several concepts, including tumour heterogeneity and the significance of GrayLevelVariance, across different sections of the text. The ensuing discourse will undertake a comparative analysis with other modalities. It is important to note that no comparison with combined clinical-radiological models is included, despite the fact that these have been shown to perform better in the literature. Clinical significance: The text does not engage with the concept of a threshold for clinical utility, for example by considering what level of AUC would be acceptable for replacing biopsy. Limitations: Despite the fact that certain subjects have been referenced, such as the utilisation of native CT and the magnitude of the sample, the exploration of selection biases, including those derived from a single cohort and the Muscovite population, should be undertaken with greater rigour. The availability of data and its capacity for replication are pivotal considerations in this field. It is indicated that data and code are not shared, which limits scientific transparency and may be viewed negatively by reviewers.

In light of the aforementioned points, I hereby propose the following amendments to be accepted following the completion of 'Major Revisions': It is imperative to broaden the scope of limitations by incorporating population bias, the absence of external validation, and the reproducibility of the pipeline. The incorporation of interobserver reliability analysis in segmentation is to be considered, or alternatively, a justification is to be provided for its non-inclusion. A review of the English language is required, with the objective of achieving a more concise and less repetitive academic style. In order to enhance the clinical credibility of the study, it is essential to incorporate interpretability, with particular emphasis on the significance of variables utilising SHAP. The following discussion will address the clinical applicability of the model, with particular reference to scenarios in which it could be used to replace biopsy. It is recommended that minimum code and data be published (anonymised features, extraction scripts) with a view to increasing impact and reproducibility. Exploration of prospective validation is warranted, encompassing multicentre and external cohort involvement.

Author Response

Dear Reviewer,

We sincerely thank you for your thorough review and valuable comments, which have greatly improved our manuscript. We have addressed all points raised and incorporated the necessary revisions. Our point-by-point responses detailing these changes are provided below.

Comment 1: “It is imperative to broaden the scope of limitations by incorporating population bias, the absence of external validation, and the reproducibility of the pipeline”.

Reply to Comment 1: We thank the reviewer for this critical insight. We have expanded the Limitations section to explicitly address population bias, the need for external validation, and pipeline reproducibility.

Changes in the text: In the Discussion section (subsection “4.4 Strengths and limitations”) added text “This work constitutes a pilot study in the Moscow population, demonstrating the feasibility of using radiomic features to differentiate LC types within the operational framework of the local radiology service. Although the Moscow population is substantial in size, the risk of population bias remains. Therefore, future research should prioritize external validation to evaluate generalizability across populations and imaging protocols. Additionally, efforts to enhance the reproducibility of the entire radiomic pipeline are essential to ensure robust and translatable results.”

Comment 2: “The incorporation of interobserver reliability analysis in segmentation is to be considered, or alternatively, a justification is to be provided for its non-inclusion.”

Reply to Comment 2: We thank the reviewer for raising this important point. We have now included a detailed interobserver reliability analysis in the Methods section, which reveals a notable difference in segmentation consistency between central and peripheral lesions and outlines the consensus-based approach used to mitigate this variability.

Changes in the text: in the Methods section (subsection "2.4.2 Details of operators performing segmentation") added text “Our sample comprised an equal distribution (50:50) of patients with central and peripheral LC types. The central lesions proved to be more challenging to delineate from surrounding tissues on native CT series alone. This was reflected in a significantly lower inter-observer agreement for central lesions, as quantified by a Dice similarity coefficient (DSC) of 0.82 ± 0.13, compared to 0.93 ± 0.07 for peripheral lesions. To mitigate this variability, a consensus-based methodology was adopted, wherein only the regions of interest (ROI) identified by both radiologists were included in the final segmentation.”

Comment 3: “A review of the English language is required, with the objective of achieving a more concise and less repetitive academic style.”

Reply to Comment 3: We thank the reviewer for this suggestion. The manuscript has undergone comprehensive editing.

Comment 4: “In order to enhance the clinical credibility of the study, it is essential to incorporate interpretability, with particular emphasis on the significance of variables utilising SHAP.”

Reply to Comment 4: We thank the reviewer for this crucial recommendation. We have now integrated a comprehensive SHAP analysis into the Results and Discussion sections.

Changes in the text: In the Results section (subsection “3.3 Feature statistics”) added text “To provide an in-depth explanation of the clinical significance of the features in the best-performing model (Gradient Boosting), SHAP analysis was performed. The beeswarm plot (Figure 5) quantified feature effects and confirmed that “GrayLevelVariance.2” was the most impactful feature, a finding consistent with the results of the afore-mentioned feature importance analysis.” and added Figure 5, representing SHAP plot for the Gradient Boosting model (the best performed one). The plot illustrates the contribution of the top predictive features, with "GrayLevelVariance.2" identified as the most influential.

In the Discussion section (subsection “4.1 Overview of important findings”) added text “According the feature importance and SHAP analysis, the most significant feature for SCLC vs NSCLC classification is the “GrayLevelVariance” of the GLDM matrix.”

Comment 5: “The following discussion will address the clinical applicability of the model, with particular reference to scenarios in which it could be used to replace biopsy.”

Reply to Comment 5: We have expanded the Discussion section to clarify information about clinical scenarios.

Changes in the text: In the Discussion section (subsection “4.2 Practical implications”) added text “Potentially, upon further validation, our model could be applied to differentiate the type of intrapulmonary tumor detected on non-contrast chest CT. This may be particularly relevant in clinical scenarios where tumor biopsy is impossible, carries high risks, or yields ambiguous cytological results. However, a formal assessment of the model's clinical utility—evaluating its net benefit over existing clinical decision pathways and its impact on patient outcomes—is a necessary next step before any routine clinical application can be considered.”

Comment 6: “It is recommended that minimum code and data be published (anonymised features, extraction scripts) with a view to increasing impact and reproducibility.”

Reply to Comment 6: Thank you for raising this important point regarding reproducibility. In response to your comment, we have made the radiomic feature dataset used in this study publicly available as a supplementary file accompanying this article. We agree that this will facilitate the validation and replication of our findings.

Changes in the text: In the Data Availability Statement section added text “The radiomic feature dataset formed and used in this study is included in this published article (and its supplementary information files). Due to institutional policies, the raw medical images, segmentation data, preprocessing scripts, source code, model files, and executable systems are not publicly available.”

Comment 7: “Exploration of prospective validation is warranted, encompassing multicentre and external cohort involvement.”

Reply to Comment 7: We have now explicitly addressed the need for prospective multicentre validation in the Limitations section.

Changes in the text: In the Discussion section (subsection “4.4 Strengths and limitations”) added text “This work constitutes a pilot study in the Moscow population, demonstrating the feasibility of using radiomic features to differentiate LC types within the operational framework of the local radiology service. Although the Moscow population is substantial in size, the risk of population bias remains. Therefore, future research should prioritize external validation to evaluate generalizability across populations and imaging protocols. Additionally, efforts to enhance the reproducibility of the entire radiomic pipeline are essential to ensure robust and translatable results.”

We thank the Reviewer for the thorough and constructive feedback, which has significantly improved the quality of our manuscript. All comments from the other reviewers have also been carefully addressed. We believe the manuscript is now substantially stronger and hope it is deemed suitable for publication.

Round 2

Reviewer 1 Report

Comments and Suggestions for Authors

The revised manuscript (jimaging-3861088) has been improved. The authors have thoroughly addressed all my previous concerns through meticulous revisions and have provided essential supplemented explanations to enhance the manuscript. Their responses are both satisfactory and well-reasoned. Accordingly, I recommend the acceptance of this manuscript for publication in Journal of Imaging.

Reviewer 2 Report

Comments and Suggestions for Authors

The study  can be accepted at this version

Reviewer 3 Report

Comments and Suggestions for Authors

Conclusion:

It is evident that a commendable effort has been made to address the majority of the 'Major Revisions' requested, resulting in a substantial improvement in the quality of the manuscript. The additions regarding the extension of limitations, interobserver reliability analysis, SHAP interpretability, and discussion of clinical applicability are particularly strong and crucial.

The primary reservations expressed pertain to the fluency of the English language in specific instances (which, although minor, could be refined for a high-impact publication) and, more crucially, the partial availability of data and code. While the intricacies of institutional policies are comprehensible, the scientific community places significant value on the complete reproducibility of research findings. The publication of the feature extraction scripts would be of significant benefit, if this could be facilitated.

In consideration of the rigour exercised during the revision process and the significant enhancement of the manuscript, it is submitted that the article is now prepared for acceptance into publication. It is hoped that future endeavours will focus on refining minor aspects of fluency and code transparency.

Congratulations on your work!